# Myosoft: An automated muscle histology analysis tool using machine learning algorithm utilizing FIJI/ImageJ software

**Lucas Encarnacion-Rivera[1,2]ʘ, Steven Foltz[1]ʘ, H. Criss Hartzell[1], Hyojung Choo**[1]*

**1** Department of Cell Biology, School of Medicine, Emory University, Atlanta, Georgia, United States of America, **2** Undergraduate program in Neuroscience and Behavioral Biology, School of Medicine, Emory University, Atlanta, Georgia, United States of America

ʘ These authors contributed equally to this work.

* hchoo2@emory.edu

**Data Availability Statement:** Myosoft is freely available to download from Github at https://github.com/Hyojung-Choo/Myosoft/tree/Myosoft-hub.

## Abstract

Skeletal muscle is comprised of a heterogeneous population of muscle fibers which can be classified by their metabolic and contractile properties (fiber "types"). Fiber type is a primary determinant of muscle function along with fiber size (cross-sectional area). The fiber type composition of a muscle responds to physiological changes like exercise and aging and is often altered in disease states. Thus, analysis of fiber size and type in histological muscle preparations is a useful method for quantifying key indicators of muscle function and for measuring responses to a variety of stimuli or stressors. These analyses are near-ubiquitous in the fields of muscle physiology and myopathy, but are most commonly performed manually, which is highly labor- and time-intensive. To offset this obstacle, we developed Myosoft, a novel method to automate morphometric and fiber type analysis in muscle sections stained with fluorescent antibodies.

### Methods

Muscle sections were stained for cell boundary (laminin) and myofiber type (myosin heavy chain isoforms). Myosoft, running in the open access software platform FIJI (ImageJ), was used to analyze myofiber size and type in transverse sections of entire gastrocnemius/soleus muscles.

### Results

Myosoft provides an accurate analysis of hundreds to thousands of muscle fibers within 25 minutes, which is >10-times faster than manual analysis. We demonstrate that Myosoft is capable of handling high-content images even when image or staining quality is suboptimal, which is a marked improvement over currently available and comparable programs.

### Conclusions

Myosoft is a reliable, accurate, high-throughput, and convenient tool to analyze high-content muscle histology. Myosoft is freely available to download from Github at https://github.com/Hyojung-Choo/Myosoft/tree/Myosoft-hub.

**Funding:** This work was supported by National Institutes of Health grants 1R01AR071397-03 (to HC), 1R01AR067786-04 and 1R01EY014852-15 (to HCH), 1F32AR074249-01A1 (to SJF), and 1R25GM125598-01 (to LER). The funders had no role in study design, data collection and analysis, decision to publish, or preparation of the manuscript.

# Background

Skeletal muscle is the most massive tissue in humans and is responsible for movement and posture [1, 2]. The human musculature comprises over 600 skeletal muscles, which generate a diverse range of contractile forces due to differences in the compositions of their constituent muscle fibers [3, 4]. Muscle fibers are broadly classified by contractile kinetics (slow or fast twitch, referred to as type I or type II, respectively). Type II fibers may be further categorized according to metabolic activity (oxidative or glycolytic) [5, 6]. According to this method of classification, there are four general fiber types: type I fibers (slow twitch/oxidative metabolism), type IIa (fast twitch/oxidative metabolism), and types IIx and IIb (faster and fastest twitch/glycolytic metabolism) [5, 7]. Each fiber type contains different myosin heavy chain (MyHC) isoforms, which differ with respect to ATPase activity and contraction speed. The contractile properties of a muscle fiber can be inferred by its size (generally reported as its cross-sectional area, CSA) and type [4, 5]. The sizes and types of fibers in a given muscle collectively contribute to its functional output [8–10].

The distributions of fiber size and type display plasticity in response to physiological pressures like aging and exercise and are altered in cases of neuromuscular disease [11–15]. Thus, analysis of fiber size and type in histological muscle preparations can be a useful method for quantifying key indicators of muscle function and for measuring responses to a variety of stimuli or stressors. However, despite the value of such analysis, it is often performed manually, which is both labor-intensive and time-consuming. To offset this obstacle, several groups have developed software that automates analysis of muscle histology [16–19]. SMASH, reported in 2014 [16], and an unnamed Image-J plug-in for muscle analysis, published in 2016 [17, 20], introduced automatic analysis of muscle histology to the field using a watershed algorithm for segmentation. Both programs, however, require manual intervention to run the program effectively, which may affect reproducibility between laboratories. Additionally, these programs lack the ability to discern all fiber types and mixed-fiber type combinations. MyoVision [18] and Muscle J [19], published in 2018, were the first programs to offer fully automated analysis of muscle histology. Myovision deploys a k-means binarization followed by detection of incompletely and completely segmented 'seeds' which the program processes independently. Although the watershed applied to incompletely segmented seeds following thresholding improves fiber detection, low-quality image inputs yield an abundance of incompletely segmented seeds, which increase the risk of false object counts. Moreover, Myovision applies these algorithms to the entire image (regardless of pixel dimensions), imposing a significant strain on the computer and making analysis of high-content images challenging. MuscleJ uses basic color transformation, Gaussian noise reduction and Li threshold binarization. This thresholding method, however, minimizes image-cross entropy, which leads to more incompletely segmented fibers. While these objects are not appropriate for analysis, they are nevertheless analyzed in MuscleJ, resulting in an overall reduction in the accuracy of the reported data. Neither Myovision nor MuscleJ permit the detection of mixed fiber types. Thus, we sought to develop a program that addressed these central limitations–specifically, the detection of all fiber types and mixed fiber type combinations and high-fidelity output even with suboptimal staining quality.

Traditional segmentation methods have relied on relatively simple mathematical transformations of images based on limited information concerning individual pixels, and the aforementioned muscle-specific programs are no exception. Unfortunately, these approaches are typically successful only when the images have high signal:noise ratios and image features are simple. Recently, machine learning algorithms have been proposed to handle digitally archived histopathological 'big data' [21]. The major advantage of data mining approaches is that they

rely on pattern recognition from minimally processed images in order to perform segmentation. Furthermore, machine learning makes it possible to distinguish between pixels that, despite sharing certain superficial similarities, do in fact belong to separate image features. This has proven useful in the context of computer-assisted diagnosis, where machine classifiers are used to assign pathological grades to biopsy samples [22, 23]. For example, a Spatially Constrained Convolutional Neural Network has been deployed for qualitative detection of nuclei in cancerous tissue [24]. Historically, this work has been performed by human experts because of the complexity of the problem: so-called histological primitives (objects like nuclei) must be identified and then further analyzed for disease relevant features (e.g. position, size/shape). Within the past decade, learning-based tools have been taken a step further: versatile and reliable methods have been developed to facilitate histological image analysis, with applications for virtually any histochemically processed sample [25, 26].

While the promise of machine learning for image segmentation has begun to be realized, it has yet to be utilized within the context of muscle biology. The key problem is one of boundary detection: fiber boundaries must first be accurately identified in order to make additional inferences about the fibers. Diverse machine-learning approaches (e.g. deep neural networks specialized for EM images or combinations of watershed algorithms and affinity-based segmentation) have already been implemented in the context of connectomics, all of which aim specifically toward boundary detection [27, 28]. Thus, we sought to train a machine-learning algorithm to detect muscle fiber boundaries in immunostained histological preparations. Recently, a machine learning based tool for image segmentation (Trainable Weka Segmentation, TWS) has been developed for the popular open-source image analysis software ImageJ, bringing the capabilities of machine learning to image analysis in biomedical research fields. In addition to increased usability of TWS as a Fiji plugin option, TWS has several advantages including freely available license, user-friendly graphic interface, and portability due to JAVA language implementation [29]. With respect to the problem of myofiber identification, a TWS classifier can be trained to recognize and distinguish between the muscle fiber boundary and intra-fiber space after a few simple manual annotations that instantiate these two features. Then, given an image, the classifier will predict the distribution of the muscle fiber boundary and intra-fiber space. The results of this prediction are represented as a probability rendering of the original image, where darker pixels represent higher probabilities for the boundary, and lighter pixels represent lower probabilities. Since the classifier is trained specifically to segment images of contiguous muscle fibers, it can "learn" to account for and clarify flaws in an immunostained image that would otherwise confound the analysis or necessitate additional segmentation methods.

Here, we present Myosoft, a novel tool to analyze muscle histology that synergizes machine learning-based image segmentation with thresholding-based object extraction and quantification. Myosoft uses pre-trained machine learning classifiers to delineate muscle fiber boundaries and subsequently extracts the size, type, and relevant morphometric features of the fiber. Additionally, Myosoft is run in the open-access image analysis software Fiji (Fiji is Just ImageJ) which is widely used to analyze cellular histology [30–32]. Altogether, Myosoft is a high-throughput, quick, accurate, and convenient solution to analyzing large sections of muscle tissue, capable of circumventing the error, bias, and labor incurred by manual annotation.

## Methods

### Mice and muscle tissue preparation

All experiments involving animals were performed in accordance with approved guidelines and ethical approval from Emory University's Institutional Animal Care and Use Committee.

C57BL/6J (n = 29) and Dmd<*mdx-4Cv*> (n = 3) mice were purchased from Jackson Laboratories. Adult mice between the ages of 3 to 9 months were used without consideration of sex. Mice were euthanized via inhalation overdose of isoflurane, the skin was removed from the hindlimbs, and gastrocnemius/soleus or tibialis anterior/extensor digitorum longus (TA/EDL) muscles were excised as a single unit. Gastrocnemius muscle tissues were mounted in OCT freezing medium (Triangle Biomedical Sciences), snap-frozen in liquid $N_2$-cooled 2-methylbutane and stored at -80°C for cryo-sectioning. Tissue cross sections of 10 μm thickness were collected every 400 μm using a Leica CM1850 cryostat.

### Immunofluorescent staining

For immunostaining specific types of myosin heavy chain and laminin, tissue sections were first treated with mouse-on-mouse reagents (M.O.M. Kit, Vector Laboratories Inc.) to block endogenous Fc receptor binding sites followed by a 1 hour incubation with 5% goat serum, 5% donkey serum, 0.5% bovine serum albumin, 0.25% Triton-X 100 in phosphate buffered saline (blocking buffer). Sections were then labeled with an undiluted 1:1:1 mixture of mouse monoclonal antibodies BA-D5 (anti-MYH7, fiber type I), SC-71 (anti-MYH2, fiber type IIa), and BF-F3 (anti-MYH4, fiber type IIb) (hybridoma supernates, Developmental Studies Hybridoma Bank) supplemented with rabbit polyclonal anti-laminin antibody (2 μg/ml, Sigma) overnight at 4°C. Control sections were incubated with species-matched non-immune IgGs. Sections were then incubated with isotype-specific Alexa Fluor (AF) conjugated secondary antibodies: anti-mouse IgG2b-AF350, anti-mouse IgG1-AF555, anti-mouse IgM-AF647, and anti-rabbit IgG-AF488 (Invitrogen, Molecular Probes) to mark type I, IIa, IIb fibers and laminin, respectively. Sections were mounted using ProLong Diamond anti-fade mountant (ThermoFisher Scientific).

### Image acquisition

All images were obtained using a Nikon Eclipse Ti-E inverted epifluorescent microscope equipped with a motorized stage. Images were acquired in NIS-Elements software (Nikon) with a 10x/0.3NA PlanFluor objective. The ND acquisition menu within Elements was used to take images from adjacent fields of view and digitally stitch them (with 15% overlap) to form a single image of the entire muscle cross-section (approximately 20–30 mm$^2$) used for analysis. For our analyses, all images were acquired as 16-bit multi-channel images, which open in Fiji as 16-bit hyperstacks. Myosoft will work with 8 or 16-bit images, but images must be formatted as hyperstacks prior to analysis. We provide an ImageJ macro for converting single channel images into a Myosoft-compatible hyperstack in S2 File.

### Image analysis

**Manual outline and fiber typing.** Four images containing 150–200 fibers and two larger images containing >500 fibers were taken from areas of the muscle section where all fiber types were represented. CSA measurement (using the polygon tool) and fiber typing (using the counting tool) was performed for all fibers in the images (excluding those on the edge) by two and three individuals, respectively, in Fiji.

**Images used to test the efficacy of Myosoft and compare it to other programs.** Images for comparison of analysis programs were generated by fractionating large, whole tissue section images. Small images (containing 200–1000 fibers) were chosen after dividing the original image into sixteenths while larger images (1000+ fibers), were taken from 2 different muscle sections divided into halves or fourths. Selection was accomplished with a random number generator to eliminate bias from the process. If an image was randomly selected and had

significant fluorescence artifacts or tissue damage, either this area was excluded, or a new image was chosen (S2 Fig). Each of these images was then run through Myosoft and its peer programs to obtain simple fiber counts. False negatives and false positives were then manually scored for each image. All above images were then classified as either poor or good quality. Stain quality was quantified as the ratio of intensity between the intra-fiber space and fiber boundary. Ratios >5 were defined as good stain quality while ratios ≤5 were defined as poor stain quality. Measurements were made at various locations to account for non-uniformity in staining within single images. The number of images that were used to generate each figure is summarized in S1 Table.

## Statistics

An unpaired Student's *t*-test was used to determine the statistical significance between two groups. The significance of differences between multiple groups was evaluated by one-way ANOVA with Bonferroni's post-test correction. Fiber size distributions were analysed using Kruskal-Wallis non-parametric ANOVA. Histogram bin sizes were determined via the Freedman-Diaconis rule: $2\frac{IQR}{n^{1/3}}$, where IQR is the interquartile range and n is the total number of observations (fibers) taken from the appropriate wildtype (WT) sample [32]. All statistical comparisons were performed using Prism 7 software (GraphPad Software, Inc). A p-value of <0.05 was considered significant.

## Myosoft download and tutorial

The code for Myosoft (an ImageJ macro) is freely available and can be accessed directly in S1 File along with a tutorial and troubleshooting instructions (S3 File) and an example image to test Myosoft (S1 Fig), or downloaded through the Choo lab repository on GitHub (https://github.com/Hyojung-Choo/Myosoft/tree/Myosoft-hub).

## Results

### Four components of the Myosoft pipeline

The Myosoft image analysis pipeline features 4 distinct modules: image pre-processing, segmentation, thresholding, and region of interest (ROI) overlay.

In the preprocessing step, the membrane-stained channel of the image is run through automatic color balancer which utilizes the histogram of intensity values to optimally enhance contrast (Fig 1 Step 1). Next, a 5x5 customized convolutional matrix is applied to the image:

$$\begin{bmatrix} -1 & -1 & -1 & -1 & -1 \\ -1 & -1 & -1 & -1 & -1 \\ -1 & -1 & 24 & -1 & -1 \\ -1 & -1 & -1 & -1 & -1 \\ -1 & -1 & -1 & -1 & 0 \end{bmatrix}$$

We find that this optimally enhances the cell boundary edges without intra-fiber noise (Fig 1 Step 2). Next, to facilitate computation on computers with limited CPU clock speed and RAM, the entire 10x muscle image is sliced into 4 to 25 (that is, $2^2$ to $5^2$; we use $4^2 = 16$) equally sized smaller images that are then processed independently (Fig 1. Step 3). Although not strictly necessary for the analysis presented here, this step drastically reduces the computing power required for execution of the Myosoft macro and is thus highly recommended (and a default setting in Myosoft).

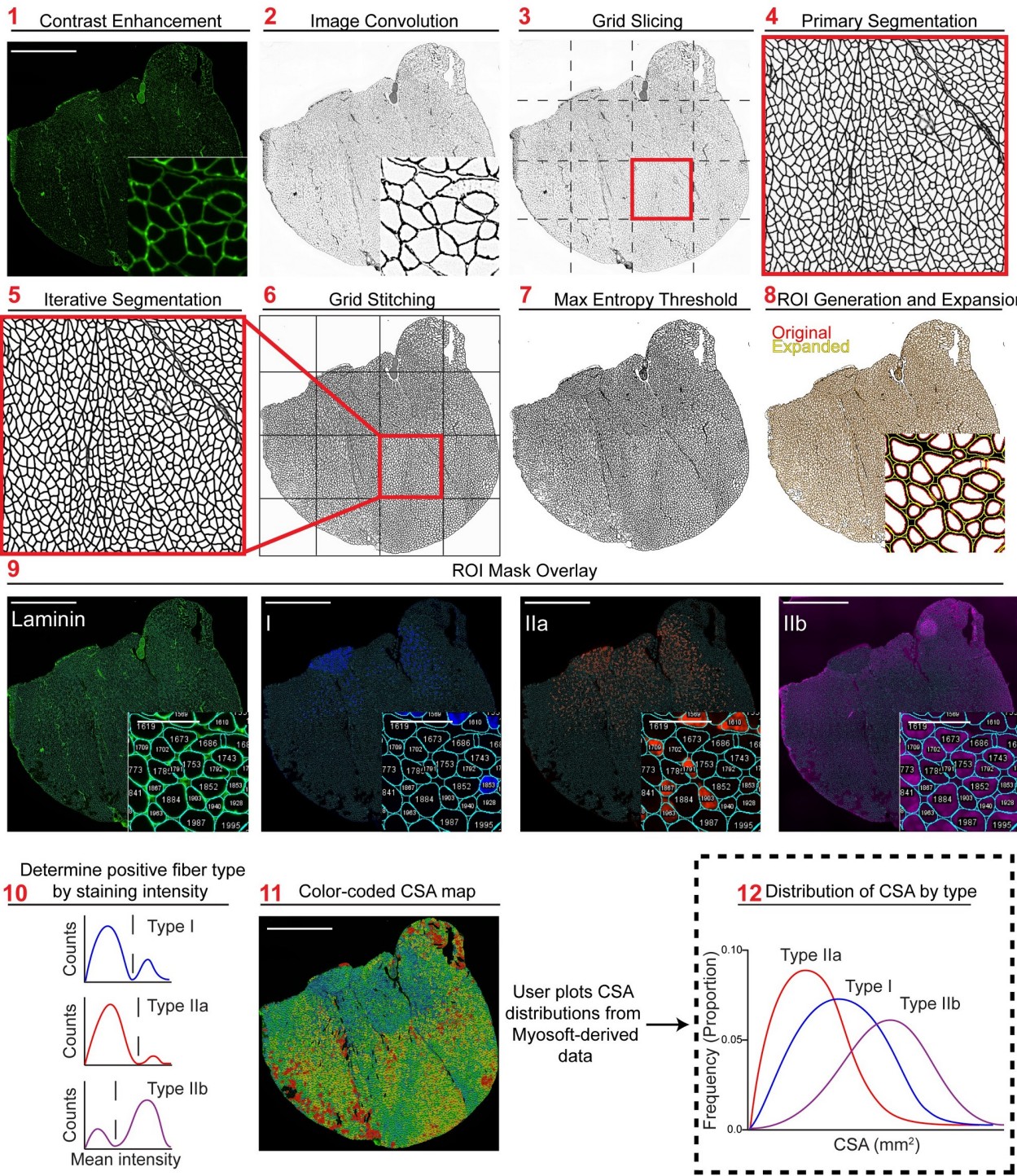

**Fig 1. Overview of Myosoft image analysis pipeline. Step 1** Image of laminin stain with contrast enhancement performed. **Step 2** Laminin stain following 8-bit conversion and image convolution. **Step 3** Image is cut in a 4x4 grid to create 16 equally sized images. **Step 4** Probability map following application of the primary machine learning classifier. **Step 5** Probability map following application of the iterative machine learning classifier. **Step 6** Recombination of all iteratively segmented grid images to form segmented full-tissue image. **Step 7** Segmented image following pixel binarization using max entropy threshold. **Step 8** Initial ROI mask (red) acquired using particle analysis and mature ROI mask (yellow) following ROI enlargement. **Step 9** Original single channel images with indexed ROI overlay, used to generate intensity histograms. **Step 10** Fiber type determination is performed by Myosoft (see Fig 2). **Step 11** Laminin stain overlaid with ROI heat map color coded by size. **Step 12** User plots CSA distributions for each fiber type.

In the segmentation step, sliced images from step 3 are segmented using a pre-trained machine learning classifier (primary classifier). The primary classifier includes difference of gaussians, Sobel, and hessian filters for feature extraction and gaussian blur and maximum filters for noise reduction. A minimum of two classes is required for training the classifier, which will then segment images into these classes. In our case, the primary classifier was trained with pixels corresponding either to the fiber boundary (high intensity pixels marked with laminin antibody) or intra-fiber space (low-intensity pixels not marked with laminin antibody). Several hundred pixels of either class were defined using the line tool in ImageJ. During classification of experimental images, each image slice is converted to a probability distribution where the intensity of each pixel (0–255) reflects the confidence of the classifier in assigning it to its respective class (Fig 1 Step 4). Next, images are subjected to another round of segmentation using a separate machine learning classifier (iterative classifier). The iterative classifier was trained to make the same classification as the primary classifier (i.e. boundary vs. non-boundary) but was trained with output images (grayscale probability maps) of the primary classifier. Both the primary and iterative classifiers are random forest models and the decision tree structure is determined by the input pixels. In this way, the output pixel generated by the decision tree of the primary classifier is run through the new decision tree structure of the iterative classifier to arrive at a new classification. The output of this step is an image with defined and continuous fiber boundaries and intra-fiber spaces with minimal noise (Fig 1. Step 5).

In the thresholding step, Myosoft retrieves all iteratively segmented images and stitches them to re-form a unified and complete segmented image of the original muscle section (Fig 1. Step 6). Next, gaussian blur is applied to the image. This blurred image is then run through a maximum entropy thresholding algorithm [33] for binarization (Fig 1 Step 7). In the binary image, pixels are either black (representing the laminin-stained cell boundary) or white (corresponding to cytoplasm of myofibers, which are unstained). A particle analyzer extracts contiguous white pixels as objects and represents them as ROIs (Fig 1 Step 8, red). The ROIs obtained after gating in step 8 are expanded according to an adjustable ROI expansion factor (Fig 1 Step 8, yellow). ROI expansion is performed to ensure that ROIs match the laminin-demarked fiber boundary as closely as possible (see laminin image inset, Fig 1 Step 9). These ROIs, which represent all fibers detected by Myosoft, are overlaid on individual channel images corresponding to each fiber type (Fig 1 Step 9). Intensity within the ROIs is measured and used to identify Type I, Type IIa, Type IIb, and mixed-type fibers (Fig 1 Step 10, Fig 2). Myosoft will generate a color-coded section map where fibers are shaded according to CSA, allowing for rapid visualization of the distribution of fiber sizes within a sample (Fig 1 Step 11). Myofiber CSAs, sorted by fiber type, are stored in .csv files readable with Excel. The user can easily plot type-specific size distributions from this data (Fig 1 Step 12). In addition to the CSA values reported in these spreadsheets, several other parameters are also reported: perimeter, circularity, minimum Feret distance, Feret angle, Feret aspect ratio, roundness, and solidity. The mean, standard deviation, minimum and maximum values for each of these measurements is also reported. Lastly, when Myosoft completes analysis, it will immediately report the fiber type proportions (for Type I, Type IIa, Type IIb, and Type IIx), total section size, and average CSA for the sample in a log window within Fiji.

## Fiber typing is determined by gating of MyHC fluorescent intensity distributions

**i. Fiber typing pipeline—Determination of IIa, IIb, I and IIx fiber types.** Following identification of myofibers within the tissue section image, Myosoft performs semi-automated

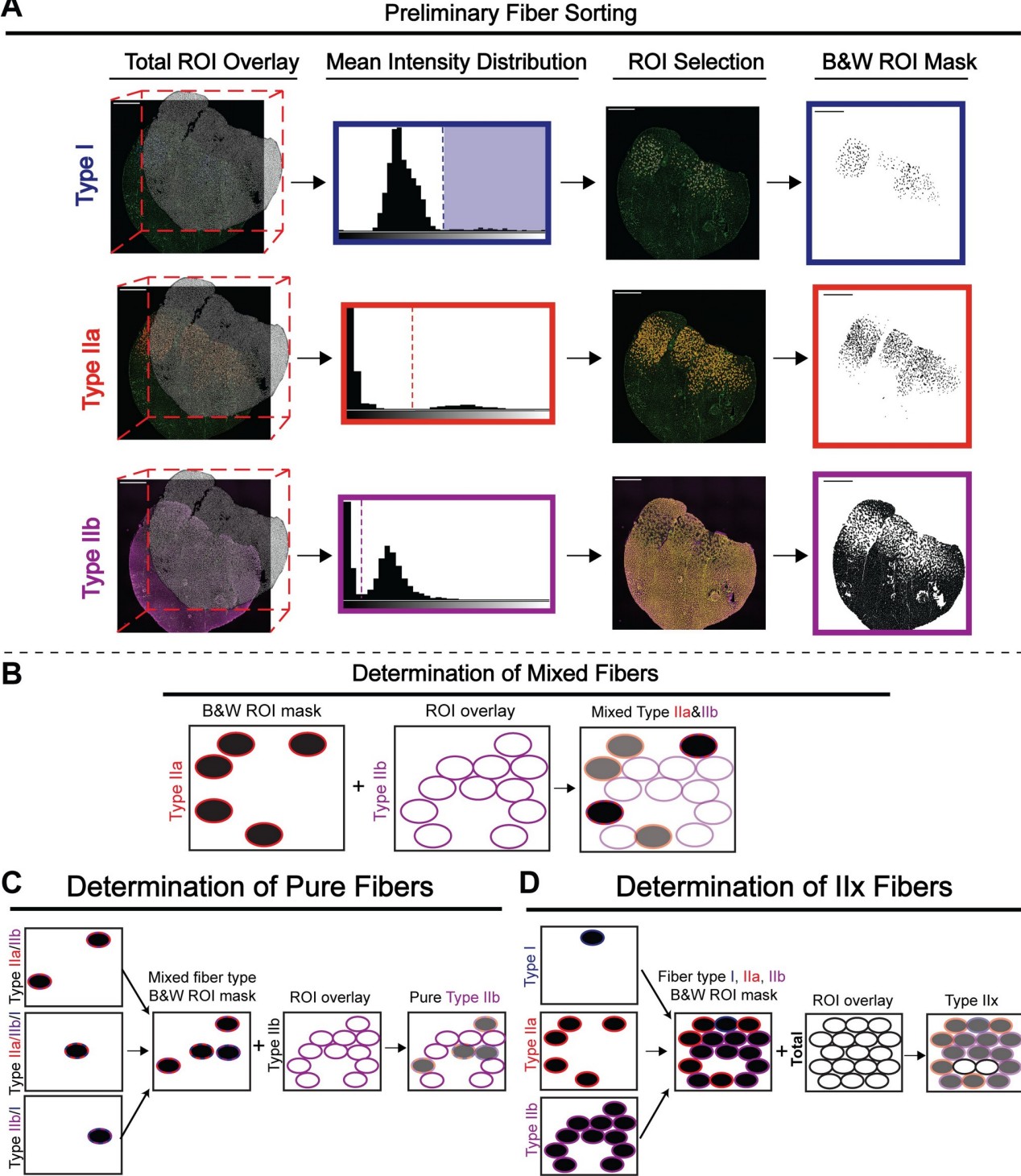

**Fig 2. Method for extracting fiber type data from Myosoft measurements. a** Intensity measurements are made on image channels corresponding to individual fiber types. These measurements are grouped into bins to generate a histogram, where low-intensity objects (background) are clustered towards the left and high-intensity objects (fibers of a given type) are clustered towards the right. The user defines a threshold intensity value (dotted line), and objects with greater mean intensity than the threshold are counted as fibers of that type. A black and white (binary) ROI mask is made for each fiber type. **b** Black and white ROI masks for each fiber type are used to identify mixed-type fibers. Identification of IIa/IIb mixed fibers is shown as an example: the black and white mask of Type IIa fibers is opened by Myosoft. Next, Type IIb ROIs are recalled, and Myosoft measures intensity within all Type IIb ROIs. Since the images are binary, the measurement values are either 0 (black) or 255 (white). Thus, modal values of 0 indicate fibers that are both Type IIa and Type IIb, or Type IIa/IIb mixed fibers. **c** Intensity thresholding (from **a**) identifies the entire population (mixed and pure/unmixed) of fibers of that type. Shown is an example of the determination of pure Type IIb myofibers. These are distinguished by first merging black

and white ROI masks of all Type IIb-containing mixed fiber types. All Type IIb ROIs are recalled, and Myosoft measures intensity within the ROIs. Mixed fibers are black (modal intensity = 0) and pure fibers are white (modal intensity = 255). **d** Type IIx fibers are identified through process of elimination. Black and white masks for Type I, Type IIa, and Type IIb fibers are merged. ROIs for all fibers are recalled, and Myosoft measures the intensity within the ROIs. Type IIx fibers are white (modal intensity = 255).

determination of IIa, IIb, and I fiber types. To accomplish this, single-channel images corresponding to each fiber type are retrieved and the total ROI mask is applied (Fig 2A. Total ROI Overlay step). Intensity values are extracted from each ROI and plotted as a frequency distribution (bin width automatically determined by Fiji). The distributions are typically bimodal, with peaks corresponding to myofibers that are positively stained for a given MyHC isoform (high intensity values at the right of the histogram) and those that are not (low intensity values at the left of the histogram). From the intensity histograms, the investigator is prompted to manually establish a threshold that will be used to define positive and negative fibers of each type (Fig 2A. Mean Intensity Distribution step). Myosoft then chooses all ROIs above the input threshold and saves the new ROI selection for each fiber type (Fig 2A. ROI selection step). Next, Myosoft recalls the ROI selection for IIa, IIb, and I fibers, recolors and fills them as black and embeds them in the image (Fig 2A. B&W ROI Mask step). Finally, to determine IIx fibers, Myosoft retrieves the total ROI mask and applies it to this black and white (BW) reference image. Myosoft then extracts the modal value from each ROI. Since type IIa, IIb, and I are black, the modal value extracted for these ROIs will be 0. Conversely, since regions that are not type IIa, IIb, nor I are empty (white), ROIs from the total mask which overlay these will possess a modal value of 255. Myosoft selects for ROIs with modal values of 255 as type IIx fibers (Fig 2D).

## ii. Fiber typing pipeline—Delineation of mixed and pure fibers

To determine mixed fiber types for all combinations (IIa/IIb, IIa/I, IIb/I, IIa/IIb/I), Myosoft retrieves each BW reference image and ROI masks from each type are successively overlaid and measured. Each time, ROIs with modal values of 0 (black)–an indication that it is a mixed fiber–are selected and saved (Fig 2B).

To find fibers that are solely IIa, IIb, or I, Myosoft retrieves the ROIs corresponding to all possible mixed combinations for each fiber type individually. For example, for IIa fibers, it will retrieve ROIs for IIa/IIb, IIa/I, and IIa/IIb/I. These ROIs are then converted into a BW mask as described above. Myosoft will then overlay the complete ROI set for that particular fiber type, as generated in Fig 2A, and will extract modal intensity values. Myosoft will then select ROIs with a modal intensity value of 255 (white) and save this as the pure ROI set for that respective fiber type. Once the above analysis has concluded, the investigator will have the ROIs and morphometric data for all fiber types. However, note that because Myosoft detects IIx fibers via the absence of other MyHC staining, it is unable to determine IIx-containing mixed fiber types (IIa/IIx and IIb/IIx). Biologically, hybrid fibers are almost always composed of MyHC isoforms that are adjacent on the spectrum of twitch kinetics (type I/IIa fibers are exceedingly more common that I/IIb fibers) [6, 34, 35]. Thus, while Myosoft will report the "presence" of type I/IIb or type I/IIa/IIb fibers, these categories are subject to errors arising from staining irregularities and the intensity thresholding method used to identify fiber types. We have chosen to maintain this aspect of the analysis pipeline because 1) these fiber types are very rarely be detected (~.5% in our test data set) and 2) counting these objects ensures that fibers are not counted more than once (Myosoft could count an erroneous "I/IIb" fiber once, or count the same fiber twice, as both a type I and a type IIb fiber). Furthermore, the legitimacy

of these fiber types may be easily verified using the Myosoft-saved ROIs and single channel images for individual fiber types.

## Adjustable parameters

In our initial tests of Myosoft, we noticed that some objects that were not myofibers were erroneously scored as myofibers. However, by visual inspection, it is clear that these objects differ in size and shape from true muscle fibers. We reasoned that exclusion criteria based on morphometric features of objects would eliminate these objects from detection. Therefore, Myosoft prompts the user to enter constraining values for several parameters that correspond to specific morphometric features. This is accomplished through the Extended Particle Analyzer plugin within the Biovoxxel Toolbox. Parameters include cross-sectional area (Fig 3A, recommended range: 50–6000μm$^2$), circularity (a measure of how well an object approximates a circle, Fig 3C, recommended range: 0.3–1.0), minimum Feret diameter (the closest possible distance between two parallel tangents of an object, Fig 3E, recommended range: 5.5–60μm), and Feret aspect ratio (the quotient of maximum and minimum Feret distances, Fig 3G, recommended range: 1–4). Adjusting these parameters can be used to exclude artifacts (e.g., interstitial spaces), objects that are not fibers (e.g. blood vessels), and improperly annotated fibers (e.g., two fibers where the boundary between them is incomplete, Fig 3B, 3D, 3F and 3H). Myosoft is pre-loaded with default (e.g. recommended) values for several morphometric parameters, which were determined experimentally through testing on a single WT control section (~6000 fibers). Parameters were adjusted one at a time and then in groups to identify conditions in which the sum of false positive and false negative measurements were minimized. It must be noted that because the recommended morphometric gates were identified on adult control tissue, it is likely that these values will need to be adjusted in certain contexts (e.g. dystrophic, regenerating, or neonatal/young muscles).

## Myosoft performance is comparable to manual analysis

We next wanted to determine how effective Myosoft is as a tool for automated myofiber size and type analysis. Two researchers with previous experience analyzing myofibers used the polygon tool in Fiji to outline muscle fibers from six section sub images (~150–600 fibers/image, ~1700 fibers total) to obtain CSA values. We then ran the same images through the Myosoft program and obtained a distribution of CSA across the images. The CSA distributions did not differ significantly between manual and Myosoft analysis (Fig 4A). Next, we tested the accuracy of fiber typing using Myosoft. Fiber type analysis was manually performed by 3 individuals on 6 images (2 images per person) representing ~3000 fibers (Fig 4B). We then used Myosoft to obtain mean intensity data for blue, red and far red channels across these four images for fiber typing. The relative proportion of each fiber type was strongly correlated between Myosoft and manual analysis ($r^2 > 0.98$) (Fig 4C).

## Myosoft is a reliable program to analyze large-scale muscle histology images

Several tools exist for automation of muscle histological analysis, but Myosoft is the first we are aware of that employs machine learning. To validate this approach, we compared the performance of Myosoft and other programs that analyze muscle histology. We chose three recently published programs for initial comparison: Myovision, SMASH, and MuscleJ [16, 18, 19]. First, we compared the muscle fiber count across programs with manual count, which is considered here to be the "true" muscle fiber count. While all muscle histology programs performed well counting myofiber number from muscle sections containing less than 500 fibers,

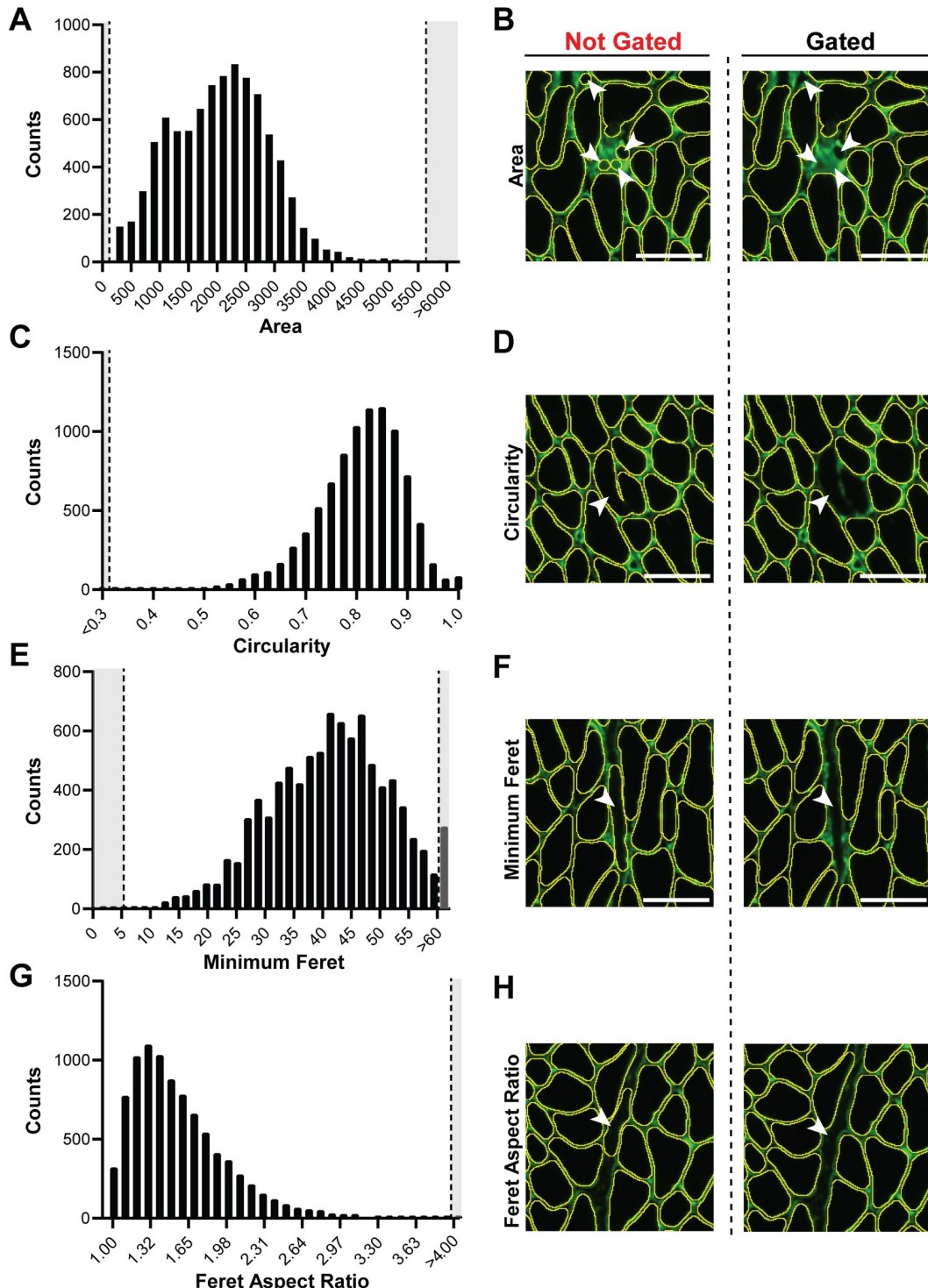

**Fig 3. Morphometric gates used to exclude false positives. a** Frequency distribution of area when no morphometric gates were used. Dotted lines indicate default threshold values used in Myosoft (this also applies to **c**, **e**, and **g**). **b** Example of non-myofiber objects excluded from analysis by area gating. Images are shown for the same tissue section where morphometric gates were not (left) or were applied (right). **c** Frequency distribution of circularity when no morphometric gates were used. **d** Example of '2 as 1' myofiber (operational classifications of false positives) excluded from analysis by circularity gating. **e** The frequency distribution of minimum Feret diameter when no morphometric gates were used. **f** Example of exclusion of endomysial space from analysis based on minimum Feret diameter gating. **g** The frequency distribution of Feret aspect ratio when no morphometric gates were used. **h** Example of endomysial space excluded by Feret aspect ratio gating. Scale bar = 50μm (all images).

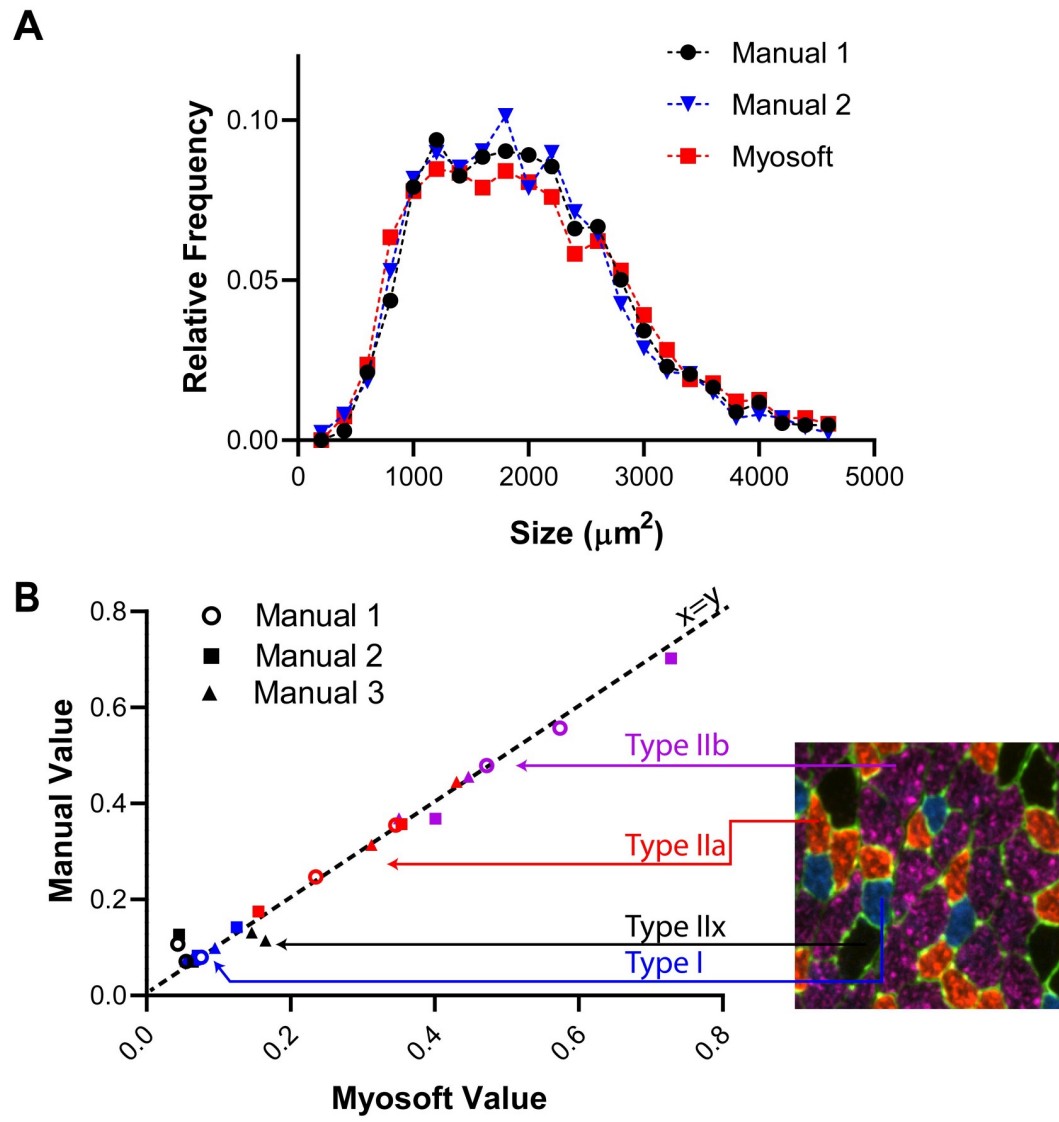

**Fig 4. Myosoft is comparable to manual analysis. a** Myofiber CSA distributions were not different when determined with Myosoft or by manual annotation (p>0.9, Kruskal-Wallis non-parametric ANOVA with Dunn's multiple comparison test). Results of manual analysis of 6 images from 2 investigators are shown. **b** Proportion of each fiber type in a given muscle section determined manually or using Myosoft. Proportions determined manually are on the y-axis and proportions determined by Myosoft are on the x-axis. Type I fibers indicated by blue symbols, Type IIa indicated by red symbols, Type IIb indicated by purple symbols, and Type IIx indicated by black symbols. Results of manual analysis of 4 images from 3 investigators are shown.

Muscle J identified fewer fibers from images containing ~750–1200 fibers, Myovision identified fewer fibers from images containing ~1200–2000 fibers, and SMASH identified more fibers from images containing ~1750–2000 fibers. Myosoft was the only program to consistently reflect manual count throughout images regardless of fiber count (Fig 5A).

Since these data only compare the raw fiber count with manual analysis, they do not represent the accuracy of the program *per se*. For example, a program may possess a high false positive and false negative rate, such that while the count may appear artificially similar to the manual count, it does not provide an accurate measure fiber number (i.e. high precision and

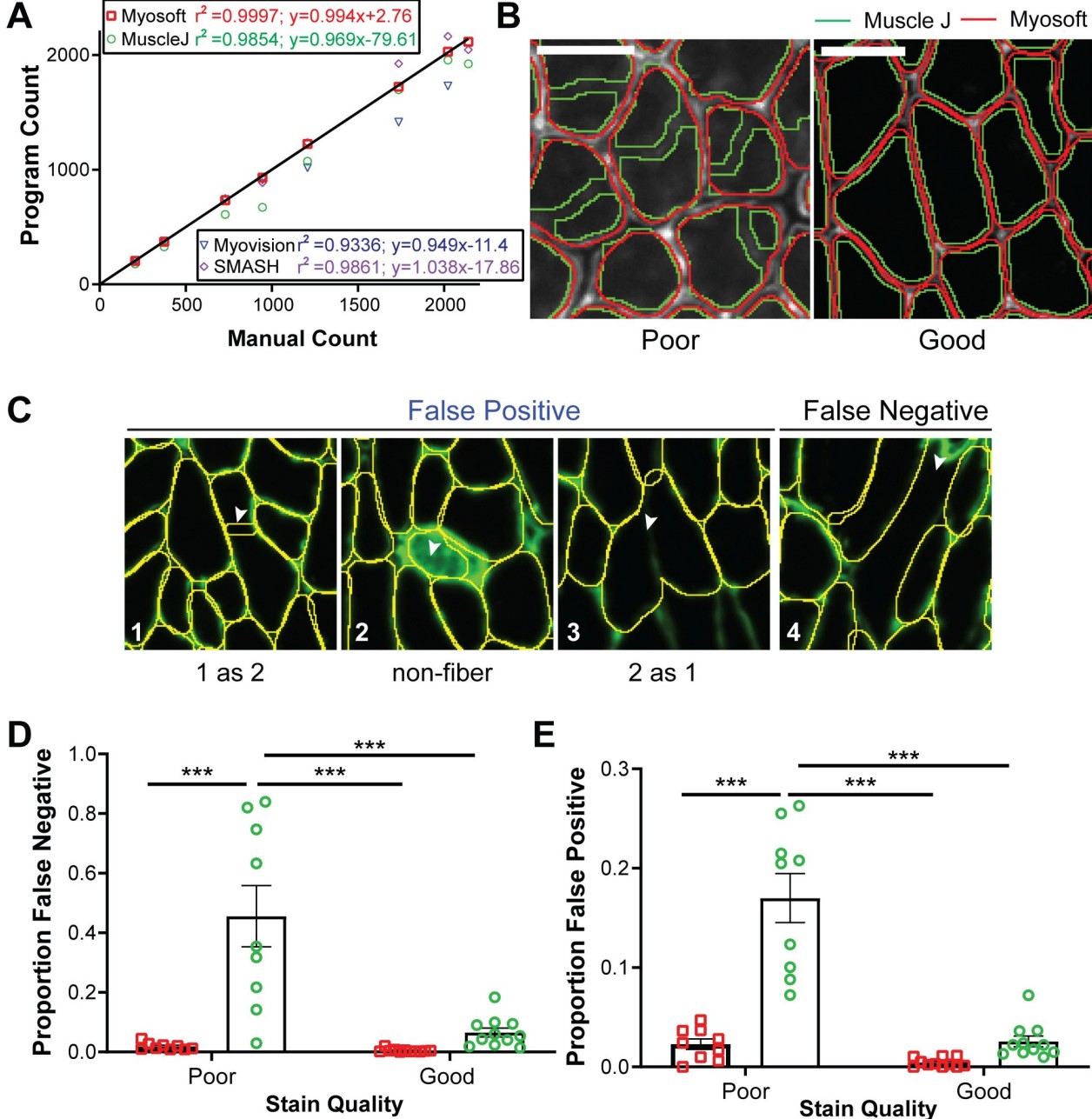

**Fig 5. Efficacy of Myosoft compared to similar programs. a** Comparison between manually counted fibers and the fiber count produced by SMASH, MuscleJ, Myovision, and Myosoft. Linear regression was performed for datasets produced by each program and equations of lines of best fit and coefficients of determination are reported in the figure key. 8 images were used to plot. **b** ROI outlines generated using Myosoft or MuscleJ in good- or poor-quality stains (scale bars = 50 μm). Good quality stains are defined as those stains which have >5-fold intensity relative to nearby un-stained space. **c** Operational classifications of false positive and negative. **d** Proportion of false negatives generated between Myosoft and MuscleJ for good- and poor-quality laminin stains. Poor staining is defined as a signal intensity (from an ROI drawn in the laminin-marked cell boundary) that is ≤5-fold greater than intensity from within the fiber. **e** Proportion of false positives generated between Myosoft and MuscleJ for good- and poor-quality stains. Each point in **c** and **d** represents analysis from a cropped region (comprising 350–800 myofibers) of a unique tissue section. 11 good-quality images and 9 poor-quality images were used. Approximately 7000 fibers were analyzed in total.

low accuracy). Further, accuracy may be more seriously affected if the quality of the stain is not optimal. Since consistently producing high quality (i.e. high contrast) stains may be impractical, utilizing a program that can retain good performance on lower quality stains is

desirable. Thus, we sought to examine Myosoft accuracy (as indicated via false positive and false negative rates) under conditions where the stain is "poor" or "good" (Fig 5B–5E). Good quality is defined as little to no noise between the muscle fiber boundary and intra-fiber space (the ratio of intensity between the intra-fiber space and fiber boundary > 5), while poor quality is defined as mid to high noise (intensity ratios ≤5) between the muscle fiber boundary and intra-fiber space (Fig 5B). False positives are defined as ROIs that do not delimit a single, whole muscle fiber (where "true" fibers are determined with manual annotation). We note three possibilities for this type of error: a single ROI outlining 2 fibers (2 as 1), 2 ROIs outlining a single fiber (1 as 2), and ROIs marking objects that are not fibers (non-fiber) (Fig 5C, 1–3). The false positive rate is defined as the number of false positives divided by the total number of fibers counted via manual analysis. Conversely, the false negative rate is defined as the rate at which the program does not generate an ROI for a fiber (Fig 5C, 4). We chose to compare Myosoft only to MuscleJ since it is the most recent muscle fiber analysis program and has the largest suite of abilities presented to date. Furthermore, it is like Myosoft in that it is capable of analyzing large-scale images, is coded in IJMacro and runs in Fiji (SMASH and Myovision require a MATLAB compiler). We chose images randomly from both "poor" and "good" quality stains (S2 Fig). When instances of false positive and false negative were counted manually for both Myosoft and MuscleJ, Myosoft displayed robust low false negative and false positive rates (<1.5%) regardless of staining quality (Fig 5D and 5E). Meanwhile, when the stain quality was poor, MuscleJ (green line in Fig 5B) performed significantly worse than Myosoft (red line in Fig 5B). With low quality images, MuscleJ's false negative rate was ~40% and false positive rate was ~17%, which were significantly greater than those of Myosoft for both good- and poor-quality stains (Fig 5D and 5E).

## Example data: Histological analysis of gastrocnemius muscles

We tested Myosoft with sections derived from WT gastrocnemius muscles (n = 8 mice). Fig 6A shows representative fiber type (top) and laminin staining images (middle). Myosoft generates a color-coded CSA map to visualize the distribution of muscle fiber size (bottom). In addition, Myosoft provides .csv files for all fibers, pure fiber types and mixed fiber types. Proportions of individual fiber types are reported to the user when Myosoft concludes analysis in an ImageJ log window, providing immediate data from the run. Mixed fiber type proportions are not reported, but are easily calculated from data given in .csv files. Proportions of all pure and mixed fiber types are shown in Fig 6B. Mixed fibers were identified at a rate of approximately 7 per 1000, indicating that although they were largely absent from our test images, Myosoft is capable of resolving them. Lastly, because Myosoft generates a unique .csv file for every possible pure or mixed type, it is simple to generate histograms of CSA values for individual fiber types. CSA distributions from our test dataset are provided in Fig 6C.

## Example: Myosoft clearly distinguishes normal and dystrophic muscle

Finally, we sought to evaluate the power of Myosoft to detect differences between normal and dystrophic tissue. For this purpose, we analyzed TA/EDL sections from WT or *mdx* mice. The *mdx* mouse is one of the most common muscular dystrophy models, harboring a spontaneous mutation in the *Dmd* gene (encoding dystrophin) and presenting with a moderate muscle phenotype [36]. *Mdx* muscle showed clear variability in its fiber size distribution, with abnormally high proportions of both atrophic and hypertrophic fibers. This feature, which is a hallmark of muscular dystrophy, was evident by inspection of the color-coded section image generated by Myosoft and through comparison of CSA histograms from *mdx* or WT muscle fibers (Fig 7A and 7C). Fiber type proportions were similar

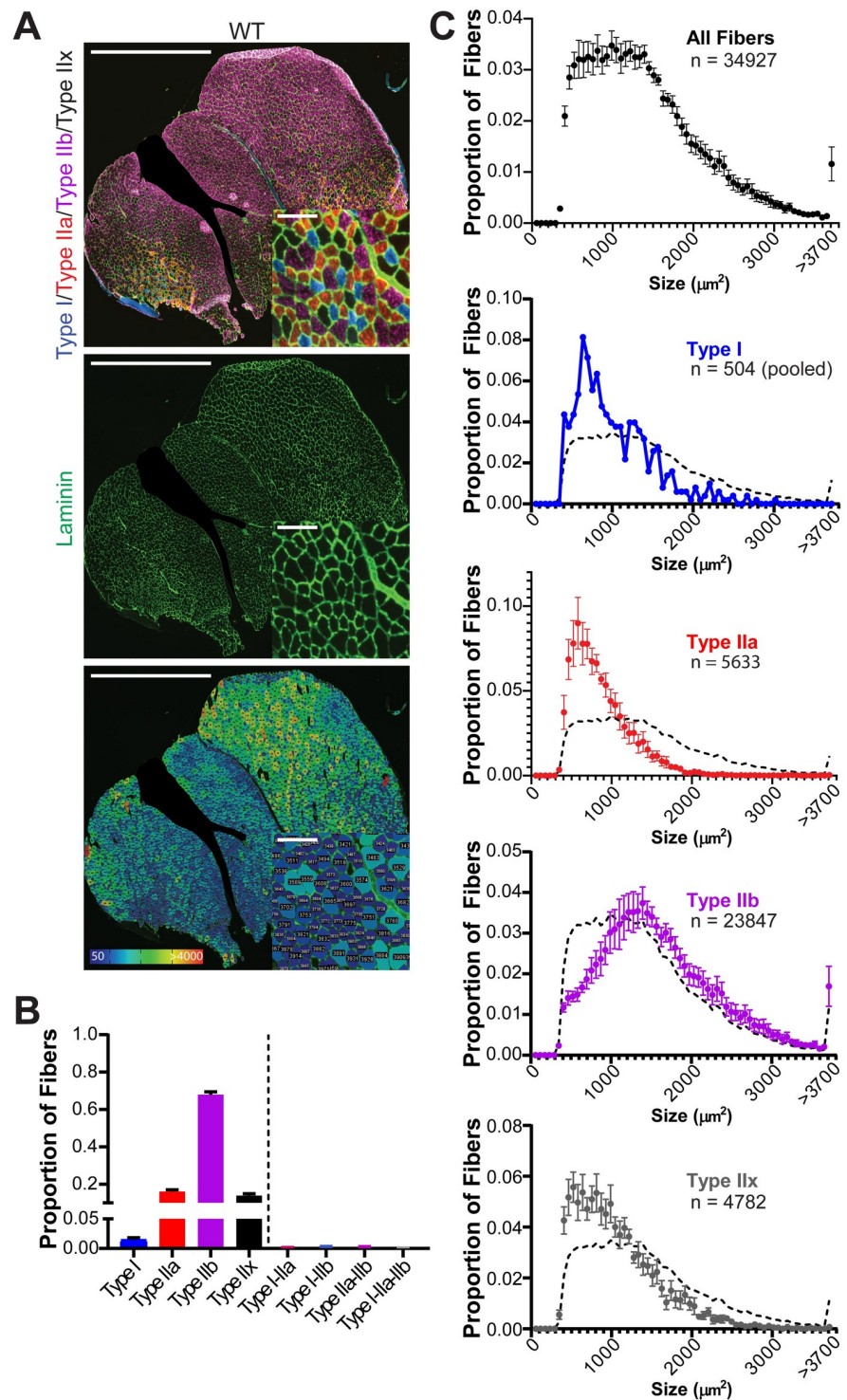

**Fig 6. Fiber-type proportions and type-specific size distributions are derived from Myosoft-generated data. a** Whole section images of WT gastrocnemius muscle showing fiber types (top), laminin, marking cell boundaries (middle), or section maps color-coded according to fiber sizes (bottom). Scale bars: 1500μm, large images; 100μm, inset images. **b** Fiber type proportions determined from analysis of gastrocnemius sections with Myosoft. Data are represented as mean ± SEM from n = 8 mice. **c** Size (CSA) distributions of myofibers in aggregate (top) or by specific fiber type. A dotted line, representing the aggregate distribution of all myofibers, is shown as a reference on all plots of type-specific size distributions. Type I fibers from 8 mice were pooled for this analysis. Large nerves were cropped out of several images, including the one shown in **a**, prior to analysis with Myosoft.

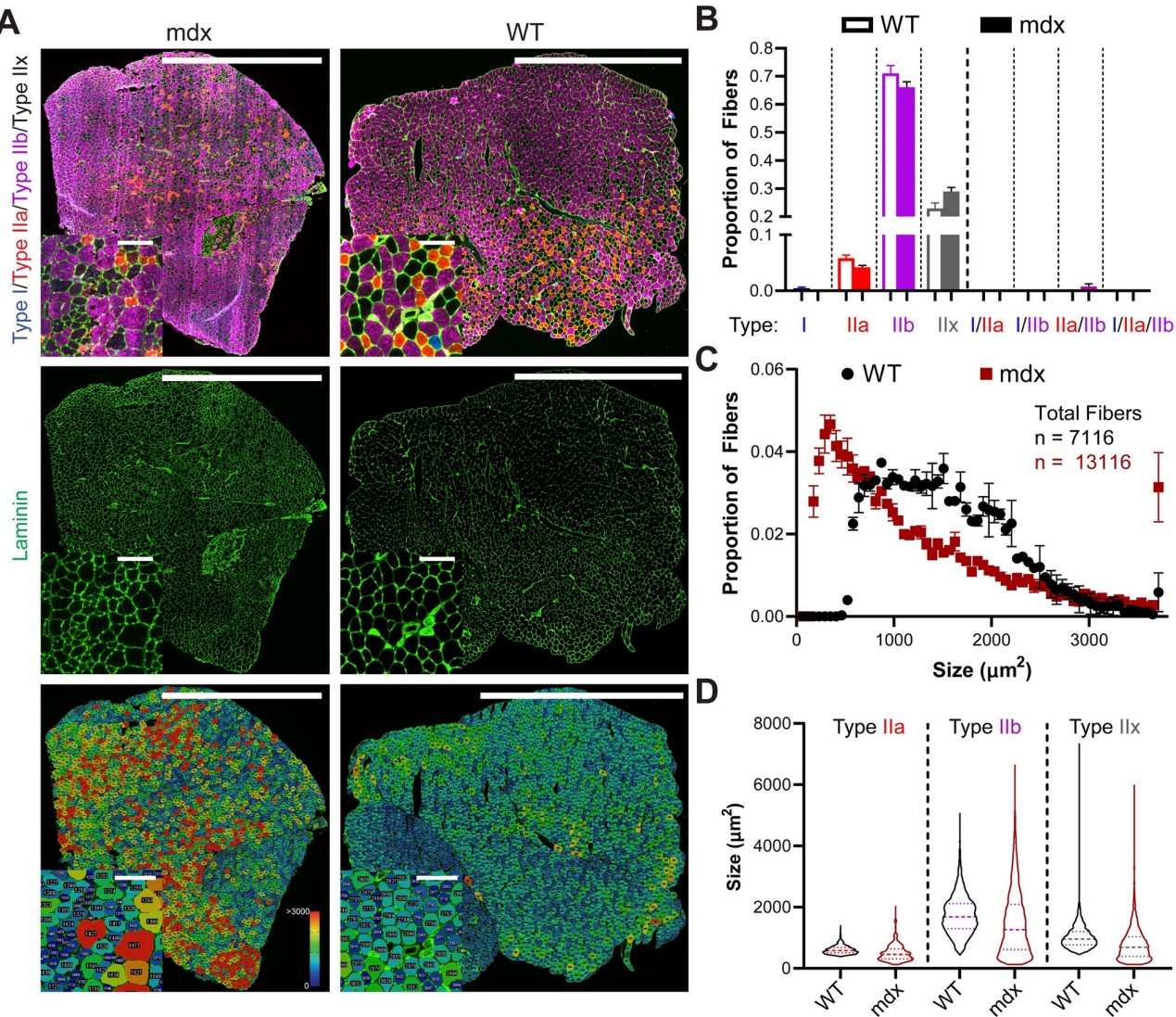

**Fig 7. Myosoft is suitable for use with dystrophic muscle sections. a** Whole section images of dystrophic (mdx, left) or control (WT, right) TA/EDL muscles showing fiber types (top), laminin, marking cell boundaries (middle), or section maps color-coded according to fiber sizes (bottom). Scale bars: 1500μm, large images; 100μm, inset images. **b** Fiber type proportions determined by analysis of TA sections with Myosoft. Data are represented as mean ± SEM from n = 3 mdx and n = 2 WT mice. **c** CSA distributions of all fibers in mdx (red squares) or WT (black circles). **d** Violin plots of individual fiber-type size distributions of WT or mdx myofibers. Distributions are approximately normal for WT fibers, while they are markedly right-skewed for mdx fibers. Dashed lines represent median values, while quartiles are indicated by dotted lines. Data are pooled for n = 3 mdx and n = 2 WT mice.

between the two groups, although *mdx* TAs tended to have more Type IIx fibers (Fig 7B). However, because the presence of IIx fibers is inferred through the absence of staining for Type I, Type IIa, or Type IIb-specific myosin heavy chain isoforms, we cannot rule out the possibility that fibers identified as IIx in this dataset are actually regenerating fibers (expressing embryonic myosin heavy chain). This is a limitation of conventional fluorescence microscopy, which uses filter cubes to distinguish light of various wavelengths. Typically, only four channels can be well resolved using this method, making it impossible to identify fiber boundaries and Type I, Type IIa, Type IIb, Type IIx and regenerating fibers.

Multispectral imaging techniques, which currently enable simultaneous imaging of up to six fluorophores, are a potential means to circumvent this issue and make it possible to identify both regenerating fibers, Type IIx fibers, and hybrid Type IIa/IIx or Type IIx/IIb fibers [37]. We expect to make future versions of Myosoft compatible with this method. Lastly, we used Myosoft to compare individual fiber-type distributions between WT and *mdx* sections. We found that distributions for every type were strongly right-skewed in *mdx* muscle (Fig 7D), which is expected for dystrophic muscle. Altogether, we provide a practical demonstration that Myosoft easily detects fiber size differences between normal and dystrophic muscle.

## Discussion

Although analyses of muscle fiber size and type are crucial in both research and clinical contexts, they are still routinely performed manually despite their laborious nature and consequently represent a substantial bottleneck for projects that require them. Recent advances in computer technology have enabled software-based automation of standard laboratory data analysis, including analysis of digital images. To date, several groups have reported tools intended for applications in histological studies of muscle, including Myovision, SMASH, and MuscleJ [16, 18, 19]. However, we found that all programs tested stumbled in analysis of large images or images with sub-optimal staining. During the development of Myosoft, additional ImageJ-based muscle analysis macros with useful features were described. First, Open-CSAM uses Huang auto-thresholding followed by particle analysis to identify fibers. As reported by the authors, although this method works well for detection of the majority of myofibers within a tissue section, it requires substantial manual supplementation (correction) to achieve high accuracy (that is, the user must individually draw ROIs for missed fibers or delete ROIs for misidentified non-fibers). Furthermore, this macro is not appropriate for fiber-type analysis [38]. The second macro, which is unnamed, is heavily dependent on achieving high signal:noise membrane counterstaining, which the authors propose to accomplish by using anti-spectrin and anti-dystrophin antibodies in tandem during their immunofluorescent processing. This is emblematic of a larger problem with automated image analysis: in general, automated analysis is successful only when staining protocols are optimized to yield higher quality (greater signal:noise) input images for subsequent analysis. If myofiber boundaries are well delineated, this macro performs well in calculating fiber CSAs, and will also segment Type I and Type II fibers. The macro also generates size-based (CSA or major/minor diameter) color-coded section maps, which, as the authors demonstrate, make it simple to visually detect differences in CSA distributions between different tissues. However, it is only built to handle two fiber types, and as such is not able to distinguish Type II subtypes from one another or identify mixed fiber types [39]. As a novel alternative, we employed a machine learning-based approach to improve the accuracy of image segmentation without altering standard protocols for tissue staining or image acquisition. By using two classifiers iteratively, Myosoft systematically improves image signal:noise without the need for manipulation of original images.

We have shown that Myosoft yields essentially equivalent fiber size distributions to manually annotated data, but this capability is not unique. A more challenging problem in muscle histology analysis is the evaluation of specific fiber type proportions and size distributions. Identification of muscle fiber types is most commonly accomplished through immunofluorescent methods, but weak labelling or low expression of certain myosin heavy chain isoforms, as well as high background fluorescence of muscle sections at certain wavelengths, can make it difficult to obtain data that is both precise and accurate. Myosoft solves this problem by storing identified fiber boundaries as ROIs and overlaying these ROIs on images of individual

fluorescent channels, each of which corresponds to a particular myosin isoform (and, by extension, fiber type). Thresholds for determining fiber types are set objectively according to the distribution of intensities for all fibers on a given channel. Although high signal:noise ratios make the task of setting the threshold simpler, we show that it is possible to determine a valid threshold even when differences between positive and negative fibers are hard to identify by eye. Furthermore, since several measurements are made for each ROI, it is possible to set thresholds based on parameters other than intensity. For example, the standard deviation of intensity may be an even more accurate (albeit less intuitive) metric for identifying fiber types because it is less sensitive to variation in staining intensity across the section or photobleaching that occurs during image acquisition.

While we focus on the use of Myosoft for fiber type analysis, it could conceivably be applied in any instance where it is necessary or desirable to make intensity measurements within identified object boundaries. For example, Myosoft is well-suited to assess muscle fusion *in vivo* with Pax7<sup>cre/ERT</sup>;tdTomato reporter mice [40]. In these mice, myofibers express tdTomato fluorescent protein following fusion with tdTomato expressing satellite cells, and more brightly fluorescent myofibers are understood to have incorporated more satellite cells (i.e. greater rate of *in vivo* fusion). Aside from this, the utility of Myosoft is not limited to intensity-based analyses. In fact, any standard measurement that can be made for an ROI can be made for a "fiber". This could be exploited to detect fibers with centrally located nuclei; the presence of a stained nucleus within the myofiber would result in a region of high fluorescent intensity surrounded by a region of low intensity and high overall variation in the intensity measurements taken for the fiber. Thus, analysis of standard deviations for mean intensity would report on the presence of centrally located nuclei. As proof of principle, we have generated a separate macro which exclusively detects central nuclei of, and we will upload this to the same repository as Myosoft on GitHub. Another application of Myosoft might be found in denervated muscle commonly features atrophic, angular fibers which could be detected by plotting distributions of morphometric features measured by Myosoft like circularity and solidity. Notably, the machine learning approach presented here is not restricted to analysis of skeletal muscle. The classifiers used in our macro could be extended to detect boundaries for non-muscle objects provided that there is sufficient distinction between the object and its boundary. Although we have not specifically tested additional applications, we posit that Myosoft could be easily modified to identify, for example, cardiomyocytes or adipocytes in histological sections so long as a suitable marker of the cell membrane is available.

Computing technology is now firmly integrated into the biological research enterprise, but rapid advances in fields such as machine learning and artificial intelligence offer new opportunities for automation of tedious analyses. Although Myosoft is, to our knowledge, the first program to exploit machine learning for use in muscle histological analysis, it is indeed only a first step. While the program represents a substantial improvement over manual analysis, it is not as complete as MuscleJ with respect to the kinds of myological analyses it will perform, such as counting vessels around myofibers and counting satellite cell number. In the future, we hope to introduce additional functionality into Myosoft so that our machine learning method can be leveraged for most or all common types of skeletal muscle histological analysis. It should also be noted that since Myosoft provides users with raw data and reference images, it is possible to corroborate results (something that is not possible within MuscleJ). We have likewise taken care to ensure that Myosoft will be simple and convenient to use for the entire muscle community through extensive beta testing and by providing detailed instructions for use/troubleshooting. Looking forward, it will be interesting to extend this approach to other types of analyses, both in muscle and beyond. As the use of automation expands in biological sciences, previously intractable research questions will become increasingly accessible.

## Conclusions

Myosoft synergizes the power of machine learning-based image segmentation with thresholding-based object extraction and quantification to obtain the morphometry and type of fibers in a given histological section of muscle. In doing so, it is capable of circumventing the time, effort, and error incurred by manual histology analysis and addresses the central limitations of its peers. Myosoft is freely available for use in the open access image analysis platform: Fiji (Fiji Is Just ImageJ), allowing access to the vast repertoire of functions therein which are familiar to much of the muscle community. Myosoft also applies the power and versatility of a machine learning-based approach to image analysis. We anticipate that Myosoft will be an especially useful tool for the muscle community and will serve as a scaffold for the creation of future automation programs.

## Supporting information

**S1 File. Myosoft macro, for use in ImageJ or Fiji.**
(IJM)

**S2 File. Hyperstack macro, for use in ImageJ or Fiji.**
(IJM)

**S3 File. Myosoft tutorial.**
(PDF)

**S1 Fig. Myosoft example image.**
(TIF)

**S2 Fig. Example images for good- and poor-quality stains and discarded sub-images.**
(TIF)

**S1 Table. Summary table of number of images for each figure.**
(DOCX)

## Acknowledgments

We appreciate Carol Zhu, Shirley Zhou and Neil Reddy for their kind efforts with manual analysis of muscle histology as well as beta-testing for Myosoft.

## Author Contributions

**Conceptualization:** Lucas Encarnacion-Rivera, Steven Foltz, Hyojung Choo.

**Data curation:** Lucas Encarnacion-Rivera, Steven Foltz, Hyojung Choo.

**Formal analysis:** Lucas Encarnacion-Rivera, Steven Foltz, H. Criss Hartzell, Hyojung Choo.

**Funding acquisition:** Lucas Encarnacion-Rivera, Steven Foltz, H. Criss Hartzell, Hyojung Choo.

**Investigation:** Lucas Encarnacion-Rivera, Steven Foltz, Hyojung Choo.

**Methodology:** Lucas Encarnacion-Rivera, Steven Foltz, Hyojung Choo.

**Resources:** H. Criss Hartzell, Hyojung Choo.

**Software:** Lucas Encarnacion-Rivera.

**Supervision:** Steven Foltz, H. Criss Hartzell, Hyojung Choo.

**Validation:** Lucas Encarnacion-Rivera, Steven Foltz, Hyojung Choo.

**Visualization:** Lucas Encarnacion-Rivera, Steven Foltz.

**Writing – original draft:** Lucas Encarnacion-Rivera, Steven Foltz, Hyojung Choo.

**Writing – review & editing:** Lucas Encarnacion-Rivera, Steven Foltz, H. Criss Hartzell, Hyojung Choo.

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
