## [Decision Letter · Decision Letter 0]

26 Sep 2019

PONE-D-19-24806

Myosoft: an automated muscle histology analysis tool using machine learning algorithm utilizing FIJI/ImageJ software.

PLOS ONE

Dear Dr. Choo,

Thank you for submitting your manuscript to PLOS ONE. After careful consideration, we feel that it has merit but does not fully meet PLOS ONE’s publication criteria as it currently stands. Therefore, we invite you to submit a revised version of the manuscript that addresses the points raised during the review process.

We would appreciate receiving your revised manuscript by Nov 10 2019 11:59PM. To enhance the reproducibility of your results, we recommend that if applicable you deposit your laboratory protocols in protocols.io, where a protocol can be assigned its own identifier (DOI) such that it can be cited independently in the future. For instructions see: http://journals.plos.org/plosone/s/submission-guidelines#loc-laboratory-protocols

We look forward to receiving your revised manuscript.

Kind regards,

Thomas Abraham, PhD

Academic Editor

PLOS ONE

Journal Requirements:

Reviewers' comments:

Reviewer's Responses to Questions

**Comments to the Author**

1. Is the manuscript technically sound, and do the data support the conclusions?

Reviewer #1: No

Reviewer #2: Partly

2. Has the statistical analysis been performed appropriately and rigorously? 

Reviewer #1: Yes

Reviewer #2: Yes

3. Have the authors made all data underlying the findings in their manuscript fully available?

Reviewer #1: Yes

Reviewer #2: No

4. Is the manuscript presented in an intelligible fashion and written in standard English?

Reviewer #1: Yes

Reviewer #2: Yes

5. Review Comments to the Author

Reviewer #1: This paper presents a semi-automated method for the morphometric and fiber type analysis in muscle sections stained with fluorescent antibodies

estimating the size and type in histological muscle preparations is useful for quantifying key indicators of muscle

function and for measuring responses to a variety of stimuli or stressors.

The method is interesting since it might help clinicians who presently manually perform this analysis; however, this is a

laborious, time consuming task, affected by inter and intra-personal variability.

Results are compared to state of the art works; however, I think that several points must be modified/extended before accepting the article.

section Introduction

The introduction is clear, the problem is well stated.

However, I think some state of the art works are missing.

More precisely, in lines 70-72, authors list some automatic segmentation techniques:

"To offset this obstacle, several groups have developed software that automates analysis of muscle histology (16-19)."

My advice is to shortly summarize the main steps of the cited methods (I believe they are the mostly related to this work).

Moreover, in subsequent lines, authors mention that learning methods are the solution to overcome staining artifacts

and they say (line 75): "Machine learning offers unprecedented potential in

76 resolving these present limitations in automated image analysis (20)."

I would firstly recall some works where machine learning methods are used to solve the problem of histochemical image segmentation.

As an example, please have a look (and I would cite) the following works, which exploit machine learning methods for stained image analysis.

[1] Madabhushi A, Lee G. (2016).

Image analysis and machine learning in digital pathology: Challenges and opportunities.

Medical image Analysis, 33: p. 170-175.

[2] Gurcan MN, Boucheron L, Can A, Madabhushi A, Rajpoot N, Yener B. (2009).

Histopathological Image Analysis: A review.

IEEE Rev Biomed Eng., 2: p. 147-171.

[1] Casiraghi E, Cossa M, Huber V, Tozzi M, Rivoltini L, Villa A, et al. (2017)

"MIAQuant, a novel system for automatic segmentation, measurement, and localization comparison of different biomarkers from serialized histological slices",

Eur J Histochem.

[2] Casiraghi E, Huber V, Frasca M, Cossa M, Tozzi M, Rivoltini L, et al. (2018).

A novel computational method for automatic segmentation, quantification and comparative analysis of immunohistochemically labeled tissue sections.

BMC BioInformatics. 19(Suppl 10): p. 357.

Next I would insert citation (20) and recall the main features of Trainable Weka Segmentation. Why authors did choose this learning

strategy, instead of others?

section Image acquisition

At the end of this section I would put a table, or anyway I would clarify:

- how many sections where acquired,

- the dimensions of the acquired sections

- how many subimages where extracted from those sections

- the dimension of the extracted subimages

- how many subimages where used for developing (and training) the system

- how many subimages where used for testing it

Please put some images to show:

- the whole section

- some of the working subimages

- some of the discarded subimages (discarded as explained in line 133-134:

"If an image was randomly selected and had significant fluorescence artifacts or tissue damage,

this area was either excluded, or a new image was chosen.")

- some good stain quality subimages (stain ratio>5)

- some poor stain quality subimages (stain ratio<=5).

A small note on line 138: I believe poor stain are less OR EQUAL TO, isn't it?

Otherwise, what about subimages with stain ratio = 5?

section Four components of the Myosoft pipeline

At first, I would insert Fig.1, the schematic drawing to recall the main step of the myosoft pipeline, at the beginning of the section and,

obviously, I would start the section with lines 185-196.

line 156: which contrast enhancement technique and which convolutional (filter) matrix are applied as preprocessing?

The latest might be a gaussian/median filter to remove

high band/salt-and-pepper, randomic noise... however please clarify.

What is the iterative classifier mentioned in line 165? Neural network, SVM, convolutional neural network, kNN, bayesian tree,...

I won't go in details with the remaining part of the algorithm described in this section, since the note is that

each step must be better explained.

All the image processing steps are just mentioned. Therefore, if a reader wants to reimplement it, he must understand the code,

which means the paper is not clear at all.

Briefly, I would suggest that authors start mentioning Fig.1 and listing the main steps of the algorithm.

Next, for each step, I would insert a subsection describing in detail the processing and the algorithms used in the step.

section Adjustable Parameters

Since the parameters are manually chosen by users, I would suggest that the author mention which procedure is used to choose the parameter values.

section Fiber typing is determined by gating of MyHC fluorescent intensity distributions

Are the thresholds automatically chosen by automatic methods (e.g. Otsu or Arbib), or are them manually set by users?

If they are manually set, what happens if any automatic thresholding or clustering method is used?

sections on Results and Discussion

These sections are well written and well presented, in my opinion, though I can't really understand the test sample size.

How many sections where used for testing?

There must be at least 50 sections to consider the work as convincing.

Comparison to existing software would be more convincing if the number of test images and mean performance ratios were reported (e.g. with a table).

Reviewer #2: In this study, the authors developed myosoft, an improved automated tool, for the analysis of muscle myofibers size and classification of fiber type populations in histological samples stained with fluorescent antibodies. Myosoft implements a machine learning approach, which appears to overcome the limitation of existing technologies that rely on the need of having a high quality staining to achieve accurate analysis. The overall work is exciting and the tool could be of great use for the biomedical field. However, considering the authors are introducing a new tool for general research use, information describing the requirements/steps for the use of this tool is necessary, as well as a fine-tuned approach for the detection of hybrid fibers.

Specific comments:

Given than the authors are introducing a new method for the histological analysis of muscle sections, a lot of details regarding the image requirements as well as the detailed steps for the use of the tool from beginning to end need to be provided. For instance:

1. There are very few details regarding the requirements for: a) image acquisition (i.e. exposure time), b) format of the images needed to be analyzed by the tool (TIFF, 8 bit, 16 bit, merged image, individual channels, etc), c) organization of the image-files for multiple sample analysis, d) any pre-treatment of the images (it was not clear whether the multiple supplementary txt files are run before the analysis), or e) the computer needs for running the program.

2. There was no tutorial with troubleshooting instructions attached as supplementary data in this manuscript or at the github site. The file attached as S2 included only a few lines of instructions with not sufficient details for a novice user. A tutorial with clear, detailed instructions of the use of the tool (preferably with images/screen shots), that includes a descriptive list of the output files obtained with an explanation of the information contained in each of them, is necessary for the correct use of the tool and an accurate interpretation of the results. Finally, it was not clear how the multiple files provided as supplementary materials should be used for the analysis of one sample (multiple samples) or for what purposes are these used. This point needs to be expanded and clarified.

3. For this tool to be pertinent to the user that requires fast information, the analysis should include an output file (or a log) that contains a summary of the following information:

a) Number of cells/µm2 (or biopsy size [mm2] so this information can be calculated),

b) Total number of cells

c) Number of cells (and %) corresponding to each fiber type

d) Average CSA of all cells (µm2)

e) Average CSA of each fiber type population (µm2)

4. It would also be important to define what is considered a sub-optimal quality staining and define the limits at which the tool can still provide an accurate analysis.

5. The authors compared myosoft with several available tools that perform similar tasks as the developed method. However, at least 2 recent publications (2019) describing automated tools using Image/J (doi.org/10.1186/s13395-018-0186-6 and doi: 10.1186/s13395-019-0200-7) have been overlooked. To have a better perspective of myosoft capabilities as compared to more recent tools, I recommend these tools should at least be mentioned in the discussion part of the manuscript.

6. One of the main issues of current tools for automated detection of fiber populations in muscle sections is the detection of hybrid fibers. Biologically fibers can adapt and switch from one fiber population to another in response to different stimuli. While I understand how you propose to quantify I/IIa fibers, I am concern about the significance of finding I/IIb or IIa/IIb or I/IIa/IIb fibers. More than a biological phenomenon, the presence of these populations together is a reflection of staining issues that needs to be addressed. More importantly is to define the criteria to select one population vs the other (i.e. in the case of I/IIb are these cells really type I (slowest) or IIb (fastest)). Finally, hybrid fibers for IIa/IIx or IIx/IIb populations cannot be accounted for with the proposed staining protocol.

Minor:

1. It would be a good reference to include the time that the tool takes to analyze a single sample (whole section)

2. Besides CSA, it is not described which information regarding fiber size (perimeter, diameters, etc) can be obtained with this program.

3. In the discussion part it is not clear whether myosoft can clearly identify and quantified central nuclei or if this is a possibility that has not been properly tested. Please expand

4. Line 97 - C57Bl /6 J or N?, also what is the number of mice used?

5. Line 274. Myovision

6. The overall quality of the images and tables in the figures, especially those in color need to be improved.

6. PLOS authors have the option to publish the peer review history of their article (what does this mean?). If published, this will include your full peer review and any attached files.

Reviewer #1: No

Reviewer #2: No

---

## [Author Response · Author response to Decision Letter 0]

2 Dec 2019

We are resubmitting our revised manuscript “Myosoft: an automated muscle histology analysis tool using machine learning algorithm utilizing FIJI/ImageJ software” (PONE-D-19-24806). We thank the reviewers for their insightful comments which have substantially improved the manuscript. We have addressed their concerns with manuscript revisions and new figures. Changes in the manuscript are indicated in red font. See below for a point by point response to their various concerns.

Reviewer #1 

This paper presents a semi-automated method for the morphometric and fiber type analysis in muscle sections stained with fluorescent antibodies estimating the size and type in histological muscle preparations is useful for quantifying key indicators of muscle function and for measuring responses to a variety of stimuli or stressors. The method is interesting since it might help clinicians who presently manually perform this analysis; however, this is a laborious, time consuming task, affected by inter and intra-personal variability. Results are compared to state of the art works; however, I think that several points must be modified/extended before accepting the article.

section Introduction

The introduction is clear, the problem is well stated. However, I think some state of the art works are missing. More precisely, in lines 70-72, authors list some automatic segmentation techniques: "To offset this obstacle, several groups have developed software that automates analysis of muscle histology (16-19)." My advice is to shortly summarize the main steps of the cited methods (I believe they are the mostly related to this work). 

We added a summary of cited methods as recommended.

Moreover, in subsequent lines, authors mention that learning methods are the solution to overcome staining artifacts and they say (line 75): "Machine learning offers unprecedented potential in 76 resolving these present limitations in automated image analysis (20)."

I would firstly recall some works where machine learning methods are used to solve the problem of histochemical image segmentation.

As an example, please have a look (and I would cite) the following works, which exploit machine learning methods for stained image analysis.

[1] Madabhushi A, Lee G. (2016). Image analysis and machine learning in digital pathology: Challenges and opportunities. Medical image Analysis, 33: p. 170-175.

[2] Gurcan MN, Boucheron L, Can A, Madabhushi A, Rajpoot N, Yener B. (2009). Histopathological Image Analysis: A review. IEEE Rev Biomed Eng., 2: p. 147-171.

[1] Casiraghi E, Cossa M, Huber V, Tozzi M, Rivoltini L, Villa A, et al. (2017)

"MIAQuant, a novel system for automatic segmentation, measurement, and localization comparison of different biomarkers from serialized histological slices", Eur J Histochem.

[2] Casiraghi E, Huber V, Frasca M, Cossa M, Tozzi M, Rivoltini L, et al. (2018). A novel computational method for automatic segmentation, quantification and comparative analysis of immunohistochemically labeled tissue sections. BMC BioInformatics. 19(Suppl 10): p. 357.

We appreciate that the reviewer provided supportive references. We added those references and mentioned in introduction section. 

Next I would insert citation (20) and recall the main features of Trainable Weka Segmentation. Why authors did choose this learning strategy, instead of others?

 Trainable Weka Segmentation (TWS) is a machine learning based tool for image segmentation, which has been developed for the popular open-source image analysis software ImageJ. In addition to increased usability of TWS by Fiji plugin option, TWS has several advantages including freely available license, user-friendly graphic interface, and portability due to JAVA language implementation (27).

section Image acquisition

At the end of this section I would put a table, or anyway I would clarify:

- how many sections where acquired,

- the dimensions of the acquired sections

- how many sub-images where extracted from those sections

- the dimension of the extracted subimages

- how many subimages where used for developing (and training) the system

- how many subimages where used for testing it

We include this information in supporting file 6.

Please put some images to show:

- the whole section

- some of the working subimages

- some of the discarded subimages (discarded as explained in line 133-134:

"If an image was randomly selected and had significant fluorescence artifacts or tissue damage,

this area was either excluded, or a new image was chosen.")

- some good stain quality subimages (stain ratio>5)

- some poor stain quality subimages (stain ratio<=5).

We include this information in supporting file 5.

A small note on line 138: I believe poor stain are less OR EQUAL TO, isn't it? Otherwise, what about subimages with stain ratio = 5?

We agree and corrected as recommended.

section Four components of the Myosoft pipeline

At first, I would insert Fig.1, the schematic drawing to recall the main step of the myosoft pipeline, at the beginning of the section and, obviously, I would start the section with lines 185-196.

We put Figure 1 at the beginning of ‘Four components of the Myosoft pipeline’.

line 156: which contrast enhancement technique and which convolutional (filter) matrix are applied as preprocessing? The latest might be a gaussian/median filter to remove high band/salt-and-pepper, randomic noise... however please clarify.

We show our custom 5X5 convolutional matrix formula to clarify.

What is the iterative classifier mentioned in line 165? Neural network, SVM, convolutional neural network, kNN, bayesian tree,...

Both the primary and iterative classifier use a multithread version of random forest. The difference is that the iterative classifier is trained using the probability distribution output of the primary classifier. Therefore, the decision-tree structure is different between the two. We provide a brief explanation of this point in the Results section (Fig. 1) where segmentation steps are described.

I won't go in details with the remaining part of the algorithm described in this section, since the note is that each step must be better explained. All the image processing steps are just mentioned. Therefore, if a reader wants to reimplement it, he must understand the code, which means the paper is not clear at all.

We added detailed information for each step.

section Adjustable Parameters

Since the parameters are manually chosen by users, I would suggest that the author mention which procedure is used to choose the parameter values.

We chose parameters that minimized the sum of the proportions of false positives and false negatives. More information about our approach has been included in the manuscript, in the Results section pertaining to Adjustable Parameters (Fig. 3).

section Fiber typing is determined by gating of MyHC fluorescent intensity distributions

Are the thresholds automatically chosen by automatic methods (e.g. Otsu or Arbib), or are them manually set by users? If they are manually set, what happens if any automatic thresholding or clustering method is used?

Originally, fiber typing required manual thresholding outside of ImageJ. We sought to maintain all functionality within ImageJ. Since, to our knowledge, there is no ImageJ plugin that outputs the intensity value where an algorithm gates (e.g. outputs the intensity value at which the inter-class variance is greatest for Otsu), we opted to include automatic plotting of the intensity distribution. We then integrated convenient GUIs which guide the investigator in choosing the proper gate (Fig 2A. Mean Intensity Distribution step). Although this adds a supervised step to Myosoft, we argue that automating and housing the fiber-typing within ImageJ streamlines the process as a whole and using automated thresholding does not add significant value to the program. 

sections on Results and Discussion

These sections are well written and well presented, in my opinion, though I can't really understand the test sample size. How many sections where used for testing? There must be at least 50 sections to consider the work as convincing.

We used 22 whole muscle section images (64 sub images) to develop Myosoft and 14 whole muscle section images (224 sub images) to test Myosoft (Supporting File 6). 

Comparison to existing software would be more convincing if the number of test images and mean performance ratios were reported (e.g. with a table).

We used 8 images to compare Myosoft and existing muscle analysis software. Fig 5A shows the performance of each software to compare manual analysis (each data point represents an image analyzed manually, or by Myosoft or the other automated methods mentioned). Overall performances are noted by formulas for lines of best fit and r2 values from regression analysis.

Reviewer #2 

In this study, the authors developed myosoft, an improved automated tool, for the analysis of muscle myofibers size and classification of fiber type populations in histological samples stained with fluorescent antibodies. Myosoft implements a machine learning approach, which appears to overcome the limitation of existing technologies that rely on the need of having a high quality staining to achieve accurate analysis. The overall work is exciting and the tool could be of great use for the biomedical field. However, considering the authors are introducing a new tool for general research use, information describing the requirements/steps for the use of this tool is necessary, as well as a fine-tuned approach for the detection of hybrid fibers.

Specific comments:

Given than the authors are introducing a new method for the histological analysis of muscle sections, a lot of details regarding the image requirements as well as the detailed steps for the use of the tool from beginning to end need to be provided. For instance:

1. There are very few details regarding the requirements for: a) image acquisition (i.e. exposure time), b) format of the images needed to be analyzed by the tool (TIFF, 8 bit, 16 bit, merged image, individual channels, etc), c) organization of the image-files for multiple sample analysis, d) any pre-treatment of the images (it was not clear whether the multiple supplementary txt files are run before the analysis), or e) the computer needs for running the program.

We added detailed information in image acquisition of Method section.

2. There was no tutorial with troubleshooting instructions attached as supplementary data in this manuscript or at the github site. The file attached as S2 included only a few lines of instructions with not sufficient details for a novice user. A tutorial with clear, detailed instructions of the use of the tool (preferably with images/screen shots), that includes a descriptive list of the output files obtained with an explanation of the information contained in each of them, is necessary for the correct use of the tool and an accurate interpretation of the results. 

We apologize for omitting the tutorial file in the last submission. We attached a detail-oriented tutorial as Supp. File 3. We also updated our Github site with newer versions of Myosoft and tutorials. 

Finally, it was not clear how the multiple files provided as supplementary materials should be used for the analysis of one sample (multiple samples) or for what purposes are these used. This point needs to be expanded and clarified. 

We described the expected data after Myosoft analysis in ‘Example data: Histology analysis of gastrocnemius muscles’ section.

3. For this tool to be pertinent to the user that requires fast information, the analysis should include an output file (or a log) that contains a summary of the following information:

a) Number of cells/µm2 (or biopsy size [mm2] so this information can be calculated),

b) Total number of cells

c) Number of cells (and %) corresponding to each fiber type

d) Average CSA of all cells (µm2)

e) Average CSA of each fiber type population (µm2)

A) through D) are provided in an ImageJ log window (pop-up) that is shown when analysis is completed. Summary information, including average and standard deviation of CSA for individual fiber types is provided in .csv files for each fiber type, provided by Myosoft.

4. It would also be important to define what is considered a sub-optimal quality staining and define the limits at which the tool can still provide an accurate analysis.

Good quality is defined as little to no noise between the muscle fiber boundary and intra-fiber space (the ratio of intensity between the intra-fiber space and fiber boundary > 5), while poor quality is defined as mid to high noise (intensity ratios < or = 5) between the muscle fiber boundary and intra-fiber space (Fig. 5B). We also provided example images for good and poor quality images in supporting file 5.

5. The authors compared Myosoft with several available tools that perform similar tasks as the developed method. However, at least 2 recent publications (2019) describing automated tools using Image/J (doi.org/10.1186/s13395-018-0186-6 and doi: 10.1186/s13395-019-0200-7) have been overlooked. To have a better perspective of Myosoft capabilities as compared to more recent tools, I recommend these tools should at least be mentioned in the discussion part of the manuscript.

We discussed recent publications in Discussion section. Briefly, Myosoft is still a superior alternative to other Image J codes due to its minimal manual input requirements and analysis of 4 kinds fiber types simultaneously.

6. One of the main issues of current tools for automated detection of fiber populations in muscle sections is the detection of hybrid fibers. Biologically fibers can adapt and switch from one fiber population to another in response to different stimuli. While I understand how you propose to quantify I/IIa fibers, I am concern about the significance of finding I/IIb or IIa/IIb or I/IIa/IIb fibers. More than a biological phenomenon, the presence of these populations together is a reflection of staining issues that needs to be addressed. More importantly is to define the criteria to select one population vs the other (i.e. in the case of I/IIb are these cells really type I (slowest) or IIb (fastest)). Finally, hybrid fibers for IIa/IIx or IIx/IIb populations cannot be accounted for with the proposed staining protocol.

We appreciate the reviewer’s concern for fiber type staining to determine mixed fibers. We realized original staining of type IIb included a portion of false positive due to high level of background signal in the blue (AlexFluor 350) channel. Since type IIb is a major population in gastrocnemius and tibialis anterior muscles, false positives from type IIb staining caused an unusual level of mixed fiber types. Therefore, we changed the fluorescence of the secondary antibody for type IIb to far red (AlexaFluor 647). With this fluorophore, type IIb fibers are more clearly delineated and mixed fiber types are detected less than 1% (Fig 6) of total fiber number. We agree that we cannot determine hybrid fiber for IIa/IIx and IIb/IIx using Myosoft, which is one of limitations of Myosoft (discussed in manuscript). 

Minor:

1. It would be a good reference to include the time that the tool takes to analyze a single sample (whole section)

We added time (25 minutes) to analyze whole gastrocnemius section (~8,000 fibers) in Abstract.

2. Besides CSA, it is not described which information regarding fiber size (perimeter, diameters, etc) can be obtained with this program.

We described in ‘Four components of Myosoft pipeline’ section in Background as follows: In addition to the CSA values reported in these spreadsheets, several other parameters are also reported: perimeter, circularity, minimum Feret distance, Feret angle, Feret aspect ratio, roundness, and solidity. The mean, standard deviation, minimum and maximum values for each of these measurements is also reported. Lastly, when Myosoft completes analysis, it will immediately report the fiber type proportions (for Type I, Type IIa, Type IIb, and Type IIx), total section size, and average CSA for the sample in a log window within Fiji. 

3. In the discussion part it is not clear whether Myosoft can clearly identify and quantified central nuclei or if this is a possibility that has not been properly tested. Please expand

The ability to detect central nuclei is an extrapolation of the principle of gating based on intensity values. The standard deviation of the intensity of a given ROI can be used to isolate centrally nucleated muscle fibers. Although we have not implemented this method in Myosoft, we have conducted preliminary tests on images with centrally nucleated myofibers as proof of principle. We have a separate macro which exclusively detects central nuclei (mentioned briefly in discussion). We will upload this to the same repository as Myosoft on GitHub 

4. Line 97 - C57Bl /6 J or N?, also what is the number of mice used?

We used C57BL/6J. We added this info in Methods.

5. Line 274. Myovision

Corrected.

6. The overall quality of the images and tables in the figures, especially those in color need to be improved.

The PDF version for review degraded the image quality due to file size limitations. The original figures are very high quality.

---

## [Editor Report · Decision Letter 1]

29 Jan 2020

Myosoft: an automated muscle histology analysis tool using machine learning algorithm utilizing FIJI/ImageJ software.

PONE-D-19-24806R1

Dear Dr. Choo,

We are pleased to inform you that your manuscript has been judged scientifically suitable for publication and will be formally accepted for publication once it complies with all outstanding technical requirements.

With kind regards,

Thomas Abraham, PhD

Academic Editor

PLOS ONE
---

## [Editor Report · Acceptance letter]

10 Feb 2020

PONE-D-19-24806R1 

Myosoft: an automated muscle histology analysis tool using machine learning algorithm utilizing FIJI/ImageJ software. 

Dear Dr. Choo:

I am pleased to inform you that your manuscript has been deemed suitable for publication in PLOS ONE. Congratulations! Your manuscript is now with our production department. 

With kind regards,

on behalf of

Dr. Thomas Abraham 

Academic Editor

PLOS ONE